# Derivation and validation of an epigenetic frailty risk score in population-based cohorts of older adults

Xiangwei Li [1,2], Thomas Delerue[3], Ben Schöttker[1,4], Bernd Holleczek[5], Eva Grill[6,7], Annette Peters[8,9,10], Melanie Waldenberger [3,8], Barbara Thorand [6] & Hermann Brenner [1,11,12]

DNA methylation (DNAm) patterns in peripheral blood have been shown to be associated with aging related health outcomes. We perform an epigenome-wide screening to identify CpGs related to frailty, defined by a frailty index (FI), in a large population-based cohort of older adults from Germany, the ESTHER study. Sixty-five CpGs are identified as frailty related methylation loci. Using LASSO regression, 20 CpGs are selected to derive a DNAm based algorithm for predicting frailty, the epigenetic frailty risk score (eFRS). The eFRS exhibits strong associations with frailty at baseline and after up to five-years of follow-up independently of established frailty risk factors. These associations are confirmed in another independent population-based cohort study, the KORA-Age study, conducted in older adults. In conclusion, we identify 65 CpGs as frailty-related loci, of which 20 CpGs are used to calculate the eFRS with predictive performance for frailty over long-term follow-up.

Frailty is defined as a state of elevated vulnerability to poor resolution of homeostasis[1]. It is frequently observed in older adult populations[2,3] and increases the risk of developing negative health outcomes including falls, physical limitations, hospitalization, and mortality[1,4,5]. A growing body of research has reported that this syndrome results from a multidimensional interplay of genetic, biological, psychosocial, and environmental factors[6–8].

Without a gold standard measurement for frailty, various models were proposed to measure and define frailty in the past years[4,9–11], and they were based on questionnaires, performance measures, routine data, or a combination of any of these. Two principal instruments were widely accepted in clinical practice[12], the performance and questionnaire-based approach suggested by Fried et al.[4] and the "Frailty index (FI)-approach" suggested by Rockwood et al.[10], which can use any kind of data as long as the included items are frailty-related and fulfill specific validated criteria. FI has been shown to be a valid predictor of morbidity and mortality in many different population-based cohorts[13,14].

Epigenetic modifications have been recognized to play a major role in aging[15,16] and aging-related conditions, such as frailty[17,18].

[1]Division of Clinical Epidemiology and Aging Research, German Cancer Research Center (DKFZ), Im Neuenheimer Feld 581, 69120 Heidelberg, Germany. [2]Medical Faculty Heidelberg, University of Heidelberg, Im Neuenheimer Feld 672, 69120 Heidelberg, Germany. [3]Research Unit Molecular Epidemiology, Institute of Epidemiology, Helmholtz Zentrum München, German Research Center for Environmental Health, D-85764 Neuherberg, Bavaria, Germany. [4]Network Aging Research, University of Heidelberg, Bergheimer Straße 20, 69115 Heidelberg, Germany. [5]Saarland Cancer Registry, Krebsregister Saarland, Neugeländstraße 9, 66117 Saarbrücken, Germany. [6]Institute for Medical Information Processing, Biometry and Epidemiology, Ludwig-Maximilians-Universität München, Munich, Germany. [7]German Center for Vertigo and Balance Disorders, Klinikum der Universität München, Munich, Germany. [8]Institute of Epidemiology, Helmholtz Zentrum München, German Research Center for Environmental Health, D-85764 Neuherberg, Bavaria, Germany. [9]Institute for Medical Informatics, Biometrics and Epidemiology, Ludwig-Maximilians-Universität München, Munich, Germany. [10]German Center for Cardiovascular Research (DZHK), Partner Site Munich Heart Alliance, Munich, Germany. [11]Division of Preventive Oncology, German Cancer Research Center (DKFZ) and National Center for Tumor Diseases (NCT), Im Neuenheimer Feld 460, 69120 Heidelberg, Germany. [12]German Cancer Consortium, German Cancer Research Center (DKFZ), Im Neuenheimer Feld 280, 69120 Heidelberg, Germany. ✉e-mail: h.brenner@dkfz-heidelberg.de

**Table 1 | Baseline characteristics of the study population**

| Characteristics | ESTHER study | | | KORA-Age study |
|---|---|---|---|---|
| | Discovery panel Subset I (n = 998) | Validation panel Subset II (n = 730) | Further validation Subset III (n = 538) | Validation in independent cohort Subset IV (n = 1010) |
| Age (years; mean ± SD) | 62.0 ± 6.7 | 61.7 ± 6.5 | 62.2 ± 6.6 | 75.9 ± 6.59 |
| Sex (N/%) | | | | |
| Men | 438 (43.9) | 322 (44.1) | 208 (38.7) | 506 (50.1) |
| Women | 560 (56.1) | 408 (55.9) | 330 (61.3) | 504 (49.9) |
| Educational levels (N/%)[a] | | | | |
| Low (≤9 years) | 747 (76.5) | 524 (73.9) | 395 (75.1) | 206 (20.4) |
| Intermediate (10–11 years) | 128 (13.1) | 112 (15.8) | 81 (15.4) | 531 (52.6) |
| High (≥12 years) | 102 (10.4) | 73 (10.3) | 50 (9.5) | 273 (27.0) |
| Body mass index (N/%)[b] | | | | |
| Underweight (<18.5 kg/m$^2$) | 5 (0.5) | 4 (0.6) | 1 (0.2) | 1 (0.1) |
| Normal weight (18.5-<25.0 kg/m$^2$) | 256 (25.7) | 187 (25.7) | 161 (29.9) | 204 (20.2) |
| Overweight (25.0–<30.0 kg/m$^2$) | 476 (47.8) | 347 (47.6) | 230 (42.8) | 495 (49.0) |
| Obesity (≥30.0 kg/m$^2$) | 258 (25.9) | 191 (26.2) | 146 (27.1) | 310 (30.7) |
| Smoking status (N/%)[c] | | | | |
| Never smoker | 510 (52.2) | 346 (48.9) | 249 (47.7) | 578 (57.2) |
| Former smoker | 312 (31.9) | 232 (32.8) | 176 (33.7) | 385 (38.1) |
| Current smoker | 155 (15.9) | 129 (18.3) | 97 (18.6) | 47 (4.7) |
| Alcohol consumption (g per day) | 9.8 ± 13.2 | 9.5 ± 12.8 | 8.9 ± 12.3 | 12.9 ± 17.4 |
| Epigenetic frailty risk score (mean ± SD) | 0.14 ± 0.02 | 0.14 ± 0.03 | 0.09 ± 0.03 | 0.12 ± 0.03 |

SD standard deviation.

[a]Data missing for 21 participants, 21 participants, and 12 participants for subset I, subset II, and subset III, respectively.

[b]Data missing for 3 participants and 1 participant for subset I and subset II, respectively.

[c]Data missing for 21 participants, 23 participants, and 16 participants for subset I, subset II, and subset III, respectively.

Recently, various DNA methylation (DNAm)-based aging algorithms have been developed[19–23] and have also been shown to be associated with frailty[24,25]. However, none of these algorithms was specifically derived for predicting frailty.

Here, we followed a previously proposed three-phase procedure[21] to derive and validate DNAm signature-based frailty risk score in a large population-based cohort study of older adults. Replication was performed in another independent population-based cohort. Frailty was defined by a frailty index (FI) based on the concept of deficit accumulation[10,26].

## Results

### Characteristics of the study populations

Table 1 shows the baseline characteristics of the participants. In ESTHER, the distributions of age, sex, body mass index, smoking status, and alcohol consumption were similar among the three subsets (all P-values > 0.05). The mean age was approximately 62 years, and a slight majority of participants were women. The majority of participants were overweight or obese (approximately 7 out of 10). Approximately half of them had never smoked, and approximately one out of six participants was still smoking at the time of enrollment. In KORA-age, the mean age was about 76 years and half of the participants were females. The education level (27% with ≥12 years of school education), body mass index levels (80% overweight or obese), and the level of alcohol consumption were higher than those in ESTHER.

Table 2 presents the distribution of FI at baseline and various follow-ups. FI and the proportion of frail participants increased with follow-up in ESTHER and KORA-age. In KORA-age, the levels of FI were higher due to the inclusion of older participants.

### Identification of frailty-related CpGs

In the discovery phase, conducted in subset I, 2220 CpGs passed the genome-wide significance threshold (FDR < 0.05). 65 CpGs located at

47 genes across 21 chromosomes were successfully replicated in the validation phase in subset II and were deemed as frailty-related methylation loci. When more comprehensively adjusted for further potential confounders using model 2, 53 CpGs were associated with FI, of which 43 CpGs were inversely associated with FI, with a decrease in FI (95% CI) per one standard deviation (SD) increase in methylation ranging from 0.92% units (0.06–1.77) to 3.07% units (1.05–5.08) (Supplementary Data 1).

Screening the literature for CpGs that were previously reported to be associated with frailty identified 15 CpGs from three studies[25,27,28]. Eight of them showed statistically significant associations with frailty in subset III (Supplementary Table 1).

### Construction of eFRS

Using LASSO regression, the number of CpGs was further reduced from 65 to 20 because many CpGs were correlated with each other. The eFRS was constructed with these 20 selected CpGs using the equation: $eFRS = 0.204 - 0.209 \times cg00921350 - 0.100 \times cg01234420 - 0.016 \times cg02867102 - 0.293 \times cg03725309 - 0.146 \times cg04955914 - 0.084 \times cg07312601 + 0.158 \times cg07349348 + 0.137 \times cg08463758 + 0.248 \times cg10408430 - 0.101 \times cg11700584 - 0.049 \times cg12510708 + 0.064 \times cg13570972 - 0.057 \times cg15058210 - 0.180 \times cg15380836 - 0.144 \times cg17860366 + 0.315 \times cg17971578 - 0.075 \times cg18791730 - 0.176 \times cg19267254 - 0.025 \times 21656937 - 0.077cg \times cg23458887$.

### Functional annotation of sets of CpGs

We performed a literature search in PubMed to obtain information on the genes that contain the 65 frailty-related CpGs (Supplementary Data 2). The 20 CpGs included in eFRS is annotated to 17 Genes. These genes are involved in various frailty-related outcomes, including different types of cancer (i.e., HDAC4, CASP9, NFE2L3, RILP, STK40, HAO2, SNX20, MRTO4, EMILIN3, P4HA3), cardiovascular disease (i.e., HDAC4,

**Table 2 | Frailty characteristics by follow-ups and subsets**

| | N (participants) | Frailty index (means ± SD) | Frailty categories [a] (N/%) | | |
|---|---|---|---|---|---|
| | | | Non-frail | Pre-frail | Frail |
| **Subset I (ESTHER study)** | | | | | |
| Baseline | 998 | 0.139 ± 0.092 | 413 (41.4) | 458 (45.9) | 127 (12.7) |
| 2-year follow-up | 966 | 0.152 ± 0.099 | 347 (35.9) | 446 (46.2) | 173 (17.9) |
| 5-year follow-up | 843 | 0.171 ± 0.107 | 253 (30.0) | 387 (45.9) | 203 (24.1) |
| 8-year follow-up | 726 | 0.198 ± 0.109 | 151 (20.8) | 349 (48.1) | 226 (31.1) |
| 11-year follow-up | 497 | 0.211 ± 0.117 | 93 (18.7) | 230 (46.3) | 174 (35.0) |
| **Subset II (ESTHER study)** | | | | | |
| Baseline | 730 | 0.143 ± 0.092 | 282 (38.6) | 339 (46.4) | 109 (14.9) |
| 2-year follow-up | 693 | 0.152 ± 0.099 | 255 (36.8) | 328 (47.3) | 110 (15.9) |
| 5-year follow-up | 618 | 0.164 ± 0.105 | 201 (32.5) | 278 (45.0) | 139 (22.5) |
| 8-year follow-up | 517 | 0.190 ± 0.108 | 117 (22.6) | 248 (48.0) | 152 (29.4) |
| 11-year follow-up | 385 | 0.199 ± 0.117 | 91 (23.6) | 164 (42.6) | 130 (33.8) |
| **Subset III (ESTHER study)** | | | | | |
| Baseline | 538 | 0.146 ± 0.096 | 206 (38.3) | 250 (46.5) | 82 (15.2) |
| 2-year follow-up | 507 | 0.157 ± 0.106 | 188 (37.1) | 212 (41.8) | 107 (21.1) |
| 5-year follow-up | 449 | 0.169 ± 0.110 | 149 (33.2) | 189 (42.1) | 111 (24.7) |
| 8-year follow-up | 369 | 0.197 ± 0.118 | 94 (25.5) | 151 (40.9) | 124 (33.6) |
| 11-year follow-up | 258 | 0.200 ± 0.125 | 64 (24.8) | 102 (39.5) | 92 (35.7) |
| **Subset IV (KORA-Age study)** | | | | | |
| Baseline | 1025 | 0.184 ± 0.120 | 236 (23.0) | 574 (56.0) | 215 (21.0) |
| 4-year follow-up | 782 | 0.217 ± 0.133 | 130 (16.6) | 401 (51.3) | 251 (32.1) |
| 8-year follow-up | 593 | 0.241 ± 0.158 | 93 (15.7) | 282 (47.6) | 218 (36.8) |

*SD* standard deviation.

[a]Non-frail: 0 < frailty index ≤ 0.100; pre-frail: 0.100 < frailty index < 0.250; frail: 0.250 ≤ frailty index.

*CASP9, SARS*), diabetes mellitus (i.e., *RPL36AL*, *SARS*), and Alzheimer's Disease (i.e., *RPL36AL*).

Supplementary Fig. 1 presents pathway enrichment and PPI network analysis of target genes of frailty-related CpGs. The enrichment heatmap (Supplementary Fig. 1A) shows that pathways of these genes include cellular macromolecule biosynthetic process, non-small cell lung cancer, viral carcinogenesis, export from the cell, and viral process. Supplementary Fig. 1B shows the relationships between these enriched terms, where each node symbolized an enriched term and a similarity > 0.3 are connected by edges. With the application of MCODE algorithm[29], three modules (regulation of kinase activity, positive regulation of kinase activity, and positive regulation of transferase activity) in the PPI network were seen (Supplementary Fig. 1C).

Supplementary Data 3 shows the results of the mQTL analysis on the CpGs included in eFRS. Altogether, we identified 3, 3, and 55 mQTLs where genetic variation was significantly associated ($P < 1 \times 10^{-7}$) with the loci cg02867102, cg07312601, and cg11700584, respectively.

**Association of eFRS with FI at baseline and each follow-up**
Figure 1 shows the correlation matrix of age, eFRS and FI at baseline and various follow-up times in the two validation subsets. In ESTHER, eFRS was more strongly related to chronological age than FI at baseline (Spearman correlation coefficients, $r_{Sp}$, 0.443 and 0.267, respectively), but the correlation of FI with age increased with increasing length of follow-up (up to $r_{Sp} = 0.397$ at 11-year follow-up). Correlation coefficients between eFRS at baseline and FI at baseline and the various follow-up times were all in the range of 0.2–0.3, and correlation coefficients between the FI at various points of time were all in the range of 0.6–0.9. Essentially very similar correlations between eFRS and FI at baseline and the various follow-ups were seen with Pearson's correlation coefficients ($r_{Sp}$ similarly ranged from 0.2 to 0.3).

Approximately 6%, 9%, 9%, 9%, and 5% variation of frailty of at baseline, 2-, 5-, 8-, and 11-year follow-up can be explained by eFRS. Furthermore, we observed a significant correlation of eFRS with AccAgeGrim ($r_{Sp} = 0.566$, $P < 0.001$). Replicated analyses in KORA-age showed consistent patterns and similar correlations. eFRS showed a slightly stronger correlation with age ($r_{Sp} = 0.427$) than FI at baseline ($r_{Sp} = 0.411$). Correlation coefficients between eFRS and FI at baseline and the two follow-ups similarly ranged between 0.2 and 0.3. About 6%, 7%, and 7% variation of frailty at baseline, 4-, and 8-year follow-up can be explained by the eFRS. All correlations in ESTHER and KORA-age were highly statistically significant ($P < 0.001$).

Supplementary Table 2 provides covariate-adjusted associations of eFRS at baseline with FI at baseline and subsequent follow-up rounds in the two validation subsets. In ESTHER, a one SD increase in eFRS was associated with an increase in the FI by approximately 2 percent units (range from 1.55 to 2.16 percent units). The associations were fairly stable across follow-up rounds and various adjustment levels, even though they did no longer reach statistical significance for FI measured at the 11-year follow-up, given the smaller number of participants who were still included in this follow-up round. In KORA-age, very similar, albeit slightly weaker associations were observed between eFRS and FI at baseline and subsequent two follow-up times (a one SD increase in eFRS was associated with 1.27-1.97 percent unit increments of FI).

Supplementary Fig. 2 illustrates multivariable-adjusted ORs (95% CIs) for the association of FRS with being pre-frail or frail at baseline and each follow-up in the two validation subsets. In ESTHER, the associations of pre-frailty/frailty with highest (vs. lowest) quartiles of eFRS were statistically significant at 2-year follow-up and 5-year follow-up with ORs 3.53 (95% CI = 1.66–7.52) and 2.86 (95% CI = 1.18–6.92), respectively. Similarly, when assessing ORs as per SD of eFRS, eFRS were strongly associated with being pre-frail or frail at baseline (OR = 1.38, 95% CI = 1.05–1.82), 2-year follow-up (OR = 1.67, 95%

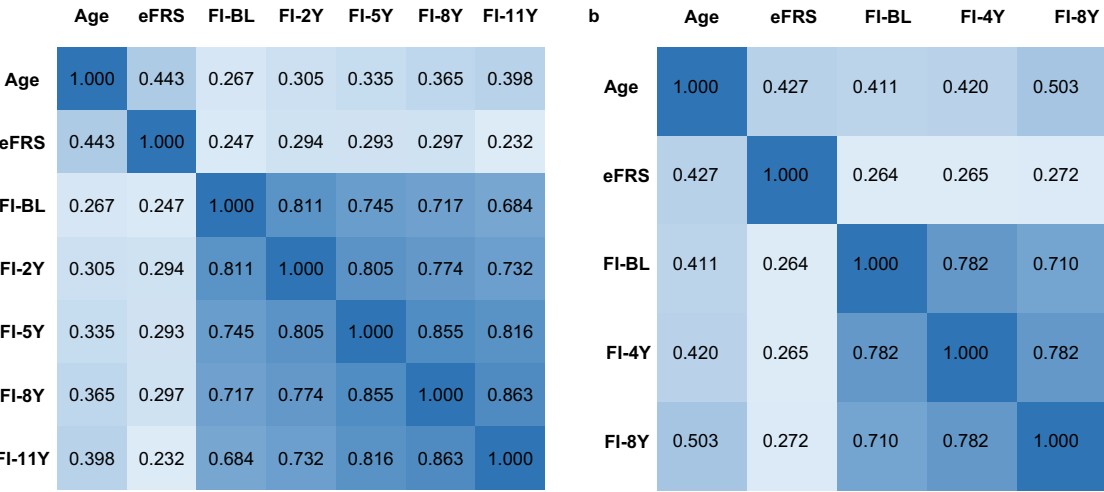

**Fig. 1 | Spearman correlation coefficients of age, epigenetic frailty risk score and frailty index at baseline and various follow-up times.** All *P*-values < 0.001 (two-sided without adjustments). **a** subset III (ESTHER study); **b** subset IV (KORA-Age study). eFRS epigenetic frailty risk score, FI-BL baseline frailty index, FI-2Y 2-year follow-up frailty index, FI-4Y 4-year follow-up frailty index, FI-5Y 5-year follow-up frailty index, FI-8Y 8-year follow-up frailty index, FI-11Y 11-year follow-up frailty index.

CI = 1.26–2.23) and 5-year follow-up (OR = 1.38, 95% CI = 1.01–1.87). Patterns were less consistent and associations were not statistically significant with pre-frailty/frailty status at 8- and 11-year follow-up. In KORA-age, strong associations of eFRS with being pre-frail or frail at baseline were seen with multivariable-adjusted ORs for eFRS quartiles 3 and 4 compared to quartile 1 as 1.99 (95% CI = 1.23–3.21) and 2.77 (95% CI = 1.61–4.75), respectively. A one SD increase of eFRS was significantly associated with a 40% increased odds of being pre-frail or frail at baseline. When restricting the analysis to those who were non-frail at baseline, similar associations were seen between eFRS at baseline and risk of being pre-frail or frail at multiple follow-ups in both ESTHER and KORA-age (Supplementary Fig. 3).

Associations of eFRS with being frail at baseline and various follow-up times in the two validation subsets are shown in Fig. 2. In ESTHER, participants in eFRS quartile 4 were at strongly increased risk of being frail at baseline (OR = 7.98, 95% CI = 2.27–28.07), 2-year follow-up (OR = 2.93, 95% CI = 1.13–7.57), and 5-year follow-up (OR = 2.92, 95% CI = 1.12–7.67), compared to participants with eFRS in quartile 1. Multivariable adjusted ORs (95% CIs) of being frail at baseline, 2-year follow-up, 5-year follow-up were 1.94 (1.31–2.89), 1.64 (1.15–2.35), and 1.48 (1.07–2.04) per one SD increase of eFRS, respectively. In KORA-age, the associations with being frail were generally weaker but statistically significant associations were still observed between eFRS and frailty at 4-year follow-up. Similar associations of eFRS with being frail at follow-ups were seen in both ESTHER and KORA-age when the analysis was restricted to participants who were non-frail or pre-frail at baseline (Fig. 3).

We further conducted sensitivity analyses with models that adjusted for smoking status using the Maas 13-CpGs model rather than self-reported smoking status in ESTHER. The positive associations were highly consistent between both types of models (Supplementary Table 3).

### Predictive performance of eFRS for frailty at baseline and follow-ups

Table 3 displays the individual and joint predictive performance of age, sex, and eFRS for being frail in the two validation subsets. In ESTHER, eFRS presented comparable predictive performances at baseline and follow-ups with the combination of age and sex. When adding FRS to models including age and sex, the predictive performance for prediction of FI at baseline and 5-year follow-up was significantly improved (from 0.629 to 0.711 at baseline and from 0.650 to 0.680 at 5-year follow-up). In the substantially older cohort of KORA-age, the

predictive performance of age and sex were generally much higher and adding eFRS to models including age and sex only slightly increased predictive performance.

### Association of AccAgeGrim with FI at baseline and each follow-up

AccAgeGrim showed similar associations with FI at baseline and subsequent follow-up times in ESTHER as eFRS (a one SD increase in AccAgeGrim was associated with 1.53–2.17% unit increments of FI, Supplementary Table 4). Similar associations of AccAgeGrim with being pre-frail or frail were also seen at baseline, 2-year follow-up, and 5-year follow-up (Supplementary Fig. 4). However, a significant relationship between AccAgeGrim with being frail was only seen at 5-year follow-up (Supplementary Fig. 5).

## Discussion

In this large-scale EWAS conducted in a population-based cohort of older adults, we identified 65 frailty-related CpGs located at 47 genes across 21 chromosomes based on DNA from whole blood, 20 of which were selected to construct the eFRS. To our knowledge, this is the first EWAS-derived eFRS, and it was found to be significantly associated with frailty at multiple points of time during long-term follow-up. These findings were validated in samples that did not overlap with samples from which the eFRS was derived and were also confirmed in an independent cohort, which demonstrated the ability of the eFRS to predict both prevalence and longer-term incidence of frailty.

To our knowledge, few studies have specifically assessed the relationship between DNAm patterns and frailty[17,18]. Only one previous EWAS on frailty defined according to the Fried criteria among 70-year-old people has been conducted and identified one CpG (cg18314882 on chromosome 8 in the *MAF1* gene)[25]. However, the single CpG is not among the CpGs identified in our study. The differences in identified CpGs within corresponding gene clusters between the two studies might be due to the different measurements for frailty. Bellizzi et al. [17] defined frailty status using cluster analysis and reported that global DNA methylation was lower in frail individuals than in the non-frail participants. This is in line with our study, which observed that 43 out of 53 CpGs were inversely associated with FI in an adjusted model. Another study by Collerton et al. [18] using the Fried frailty definition in a cohort with 85 years old participants found that the genome-wide methylation was not associated with frailty status. Recently, several DNAm-based algorithms, i.e. Hannum's blood-specific clock[19] and

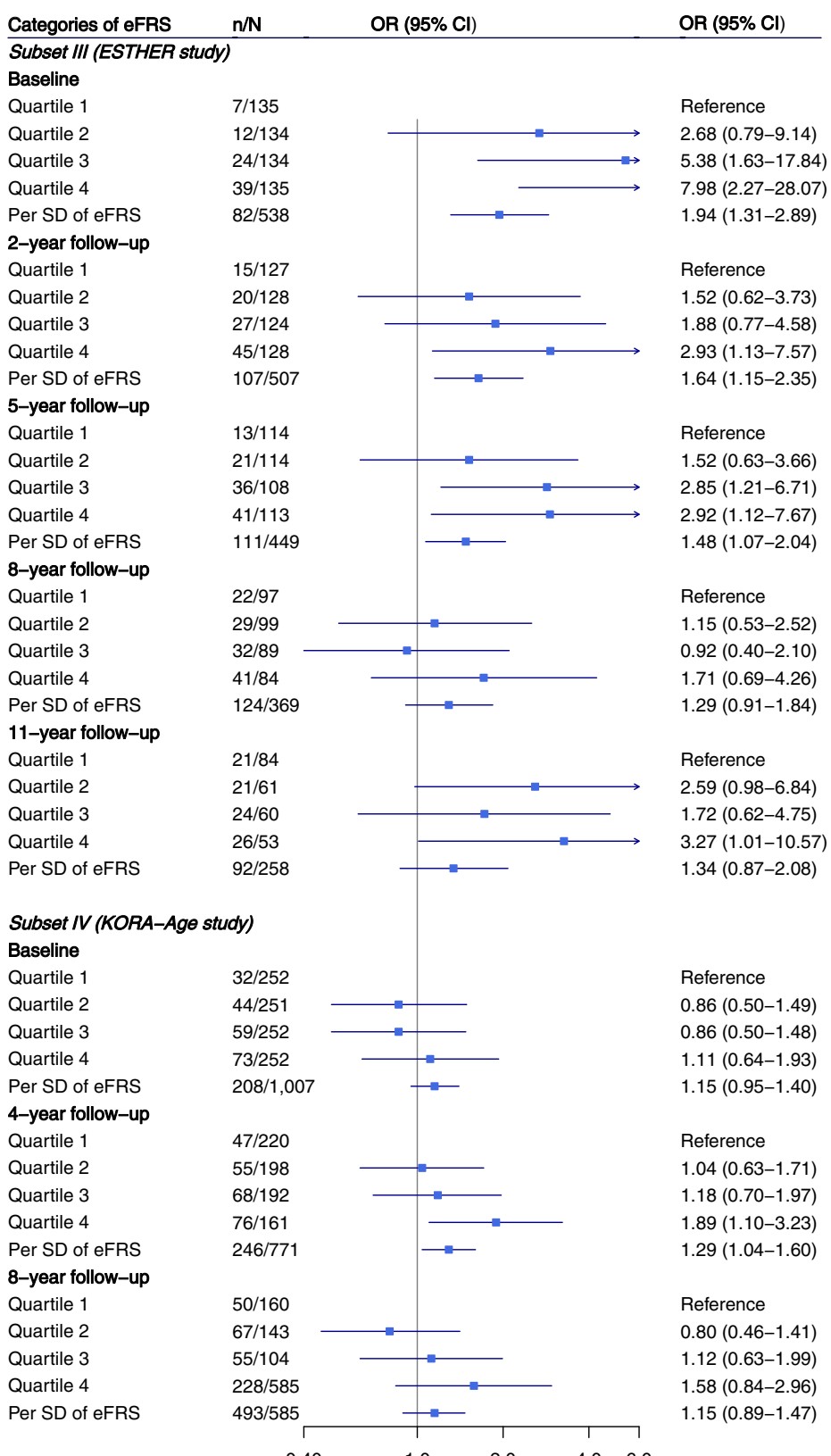

**Fig. 2 | Association of epigenetic frailty risk score with being frail at follow-ups.** Vertical ticks within the blue boxes and horizontal lines show the OR and 95% CI. Models were adjusted for age, sex, leukocyte composition, batch, baseline smoking status (never smoker, former smoker, current smoker), and alcohol consumption (g per day). eFRS epigenetic frailty risk score, OR odds ratio, CI confidence interval, SD standard deviation.

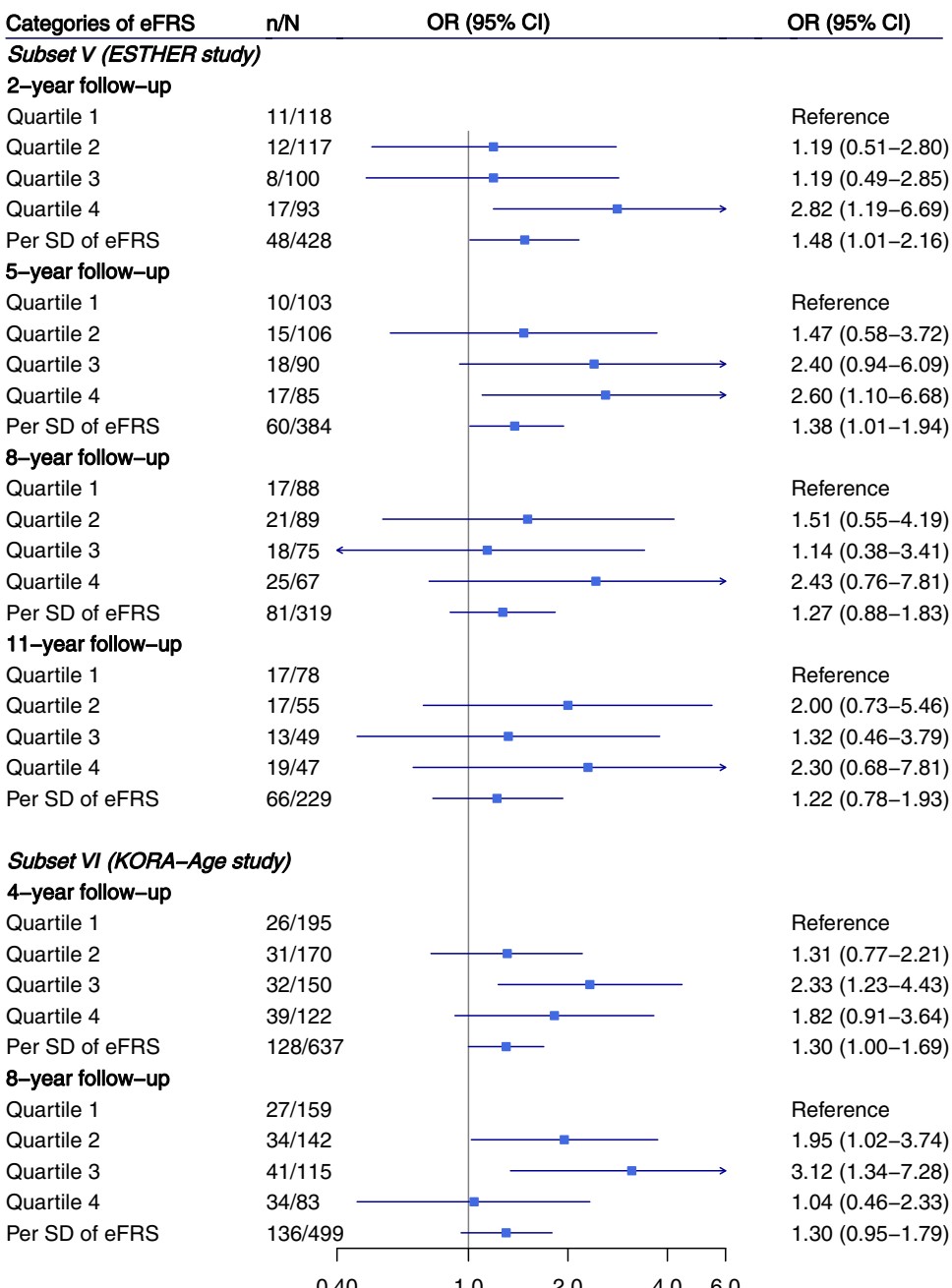

**Fig. 3 | Association of epigenetic frailty risk score with being frail among participants who were being non-frail or pre-frail at baseline.** Vertical ticks within the blue boxes and horizontal lines show the OR and 95% CI. Models were adjusted for age, sex, leukocyte composition, batch, baseline smoking status (never smoker, former smoker, current smoker), and alcohol consumption (g per day). eFRS epigenetic frailty risk score, OR odds ratio, CI confidence interval, SD standard deviation.

GrimAge[23], have been derived to estimate 'epigenetic age' and were suggested to closely correlate with frailty[24,25,30].

The identified frailty-related CpGs highlight several genes or genetic regions and might promote further investigation of the biological mechanism of frailty. Glycolytic glyceraldehyde-3-phosphate dehydrogenase (*GAPDH*) contains two of the frailty-related CpGs (cg00252813 and cg02519286) and has been found to be likely related to the pathogeneses of amyotrophic lateral sclerosis and Huntington's disease[31,32], various forms of cancers[33], and neurodegenerative disorders[33]. Three CpGs (cg01406381, cg21766592, and cg25607249) are located at Solute Carrier Family 1 Member 5 (SLC1A5), a high-affinity L-glutamine transporter that was highly expressed in several cancer types[34–36]. The remaining CpGs and correspondent genes, such

as *SARS*, *SCRN1*, *PIK3CD*, *RUNX1*, *NCAPH*, *HDAC4*, and *VAC14*, were also observed to be associated with multiple diseases. The roles of the identified CpGs in the development and/or progression of diseases may explain the association of eFRS with frailty.

We derived a DNAm-based algorithm for predicting frailty, measured by the FI, and this eFRS showed robust predictive performance for frailty in a comprehensive validation chain. Of the 20 CpGs included in the eFRS, 8 CpGs map to intergenic regions with unknown function, and the other 12 CpGs are annotated to genes involved in common chronic diseases, including coronary artery disease, stroke, type 2 diabetes mellitus, and multiple types of cancers[37–40]. Frailty has been shown to be strongly associated with a broad range of adverse health outcomes, such as worsening mobility[4], falls[38], fracture[41], and

**Table 3 | AUC (95% CI) of chronological age, sex, and epigenetic frailty risk score in prediction of being frail at baseline and each follow-up**

| Categories | AUC (95% CI) | P [a] |
|---|---|---|
| **Subset III (ESTHER study)** | | |
| **F-BL** | | |
| Age, sex | 0.629 (0.566–0.692) | |
| eFRS | 0.702 (0.641–0.763) | |
| Age, sex, eFRS | 0.711 (0.652–0.770) | 0.01 |
| **F-2Y** | | |
| Age, sex | 0.644 (0.584–0.704) | |
| eFRS | 0.666 (0.607–0.724) | |
| Age, sex, eFRS | 0.682 (0.625–0.739) | 0.07 |
| **F-5Y** | | |
| Age, sex | 0.650 (0.591–0.708) | |
| eFRS | 0.658 (0.602–0.714) | |
| Age, sex, eFRS | 0.680 (0.624–0.736) | 0.04 |
| **F-8Y** | | |
| Age, sex | 0.661 (0.601–0.720) | |
| eFRS | 0.631 (0.571–0.691) | |
| Age, sex, eFRS | 0.671 (0.613–0.729) | 0.31 |
| **F-11Y** | | |
| Age, sex | 0.689 (0.621–0.756) | |
| eFRS | 0.595 (0.522–0.668) | |
| Age, sex, eFRS | 0.691 (0.624–0.758) | 0.38 |
| **Subset IV (KORA-Age study)** | | |
| **F-BL** | | |
| Age, sex | 0.741 (0.702–0.780) | |
| eFRS | 0.620 (0.579–0.662) | |
| Age, sex, eFRS | 0.745 (0.706–0.784) | 0.23 |
| **F-4Y** | | |
| Age, sex | 0.715 (0.676–0.755) | |
| eFRS | 0.630 (0.588–0.672) | |
| Age, sex, eFRS | 0.720 (0.681–0.760) | 0.03 |
| **F-8Y** | | |
| Age, sex | 0.759 (0.717–0.801) | |
| eFRS | 0.638 (0.592–0.685) | |
| Age, sex, eFRS | 0.763 (0.721–0.804) | 0.44 |

*AUC* area under the curve, *eFRS* epigenetic frailty risk score, *F-BL* baseline frailty status, *F-2Y* 2-year follow-up frailty status, *F-5Y* 5-year follow-up frailty status, *F-8Y* 8-year follow-up frailty status, *F-11Y* 11-year follow-up frailty status.
[a]*P*-value for adding eFRS to the model containing age and sex (two-sided *P*-value with adjustments).

mortality[42]. The shared linkage with morbidity may therefore explain the association between eFRS and frailty. Moreover, the associations of 12 CpGs and common chronic diseases also support the potential capacity of eFRS for predicting adverse health outcomes and even fatal outcomes. Given that frailty may potentially be preventable up to some possible point of no return, the potential predictive performance of eFRS might be helpful for designing, implementing, and evaluating interventions aimed to prevent or slow down the development of frailty.

Several accurate composite algorithms of chronological age or lifespan have been built based on DNAm, including the Mortality Risk Score (MRscore) by Zhang et al.[21], PhenoAge by Levine et al.[22], and GrimAge by Lu et al.[23]. These DNAm-based algorithms were demonstrated to be robust predictors of mortality, lifespan, and healthspan[21–23]. In our study, a high correlation between eFRS and GrimAge was observed. Such correlation is not unexpected given that

GrimAge includes DNAm-based surrogate biomarkers for health-related plasma proteins and smoking pack-years[23] and the majority of the CpGs used to construct eFRS are annotated to health-related conditions. When comparing CpGs included in eFRS and these aging algorithms, three overlapped with GrimAge (cg02867102, cg07312601, and cg11700584). Moreover, cg02867102 and cg11700584 were significant signals in previous EWASs on smoking and aging[43–45], which points to the major role of smoking in adverse health outcomes in old age including frailty.

The current study has several strengths including the large-scale random samples from the general population, the long-term follow-up with repeated measurements of frailty at multiple points of time during follow-up, and replication in a completely independent cohort. However, it is also necessary to consider the limitations of the present study when interpreting the results. First, the deficits for the measurement of FI were self-reported which might have led to potential reporting bias. However, in the ESTHER cohort, self-reported diseases which account for a large share of the self-reported deficits were found to be in high agreement with medical records in careful validation steps carried out at each follow-up. Second, the recruitment of the participants was conducted during a voluntary health check-up. Therefore, the participants of the ESTHER cohort might not be a fully representative sample of the general population. Nevertheless, the prevalence of risk factors and chronic diseases has been found to be comparable to those observed in the corresponding age range in a representative health survey from Germany which took place at the same time as ESTHER baseline recruitment did[46]. Third, in the discovery phase, only the CpGs included in both the 450K and EPIC array were included in the EWAS. Potential frailty-related CpGs exclusively covered by the EPIC array might have been missed.

In conclusion, in this EWAS on frailty conducted in a large population-based cohort, we identified 65 frailty-related CpGs and derived an epigenetic algorithm of frailty based on 20 CpGs. The DNAm-based eFRS was demonstrated to be strongly associated with long-term frailty and validated both internally in an independent subset and in an external population-based cohort. Further studies should investigate the associations of eFRS with additional health outcomes and its potential use for earlier detection of frailty risk and designing, monitoring, and evaluating prevention measures.

## Methods
### Study population and study design
The epigenome-wide association study (EWAS), including derivation and internal validation, is based on the ESTHER study, an ongoing prospective, population-based cohort study of older adults conducted in the federal state of Saarland, Germany. Details of the study design and population have been reported previously[21,47]. Briefly, men and women aged 50–75 undergoing a general health check-up in Saarland, a small federal state in southwestern Germany, from 2000 to 2002 were eligible for participation. At the time of recruitment, these general health exams were routinely offered every 2 years to people aged 35 years and older by their general practitioners (GPs). Overall, 9940 adults aged 50–75 years were recruited by their GPs and were followed by participant and GP questionnaires after 2, 5, 8, and 11 years During baseline enrollment and each follow-up, standardized questionnaires for participants and their GPs were used to collect extensive basic data on sociodemographic characteristics, risk factors, lifestyle factors, and medical history. Moreover, whole blood samples were collected at baseline from which DNA was extracted. The ESTHER cohort was found to be representative of the older German population with respect to major sociodemographic, lifestyle, and medical characteristics[46].

Three subsets were randomly selected in three different rounds of methylation analysis from the ESTHER cohort for epigenome-wide

DNA methylation data measurements. Subsets I and II included 998 and 741 randomly selected subjects for whom DNAm measurements were performed in August 2018 and July 2019 for various projects[47,48]. Subset III has a nested case-control design for mortality-related methylation signatures and 548 participants were randomly selected as the subcohort irrespective of death status[21]. Subset I was used as a discovery panel in the epigenome-wide screening for CpG sites related to frailty at baseline. Subset II was utilized as the first internal validation panel to further select CpG sites to construct a methylation-based epigenetic frailty risk score (eFRS). The associations of eFRS with FI at baseline and during 11 years of follow-up were further validated in subset III.

Replication in an independent cohort was performed in the KORA-Age study, a population-based cohort study conducted in the region of Augsburg, Southern Germany, whose study population and design have been described in detail previously[49]. In 2008 and 2009, 4123 participants aged ≥65 years from four population representative surveys conducted between 1984 and 2001 were enrolled and completed the baseline assessments. Then, an age- and sex-stratified random sub-sample ($n = 1079$) additionally completed medical examinations and was followed up in 2012 and 2016. Methylation data were available for 1010 participants from this random sub-samples (subset IV in the current analysis). The baseline, 4- and 8-year follow-up data of these 1010 participants were used to validate the associations of eFRS with FI in an independent cohort.

The ESTHER study was approved by the ethics committees of the medical faculty of Heidelberg University and of the medical board of the state of Saarland. The KORA-Age study was approved by the Ethics Committee of the Bavarian Medical Association (EK No. 08064). All ESTHER and KORA-Age participants provided written informed consent.

### DNA methylation assessment
DNAm profiles of subsets I and II from ESTHER, and subset IV from KORA-Age were assessed with the Infinium Methylation EPIC BeadChip kit (EPIC, Illumina, Inc., San Diego, CA, USA), and DNAm profiles of subset III from ESTHER were determined with the earlier introduced Infinium Human Methylation450K BeadChip Assay (450K, Illumina, Inc., San Diego, CA, USA). Details of the methylation analysis in the ESTHER study have been reported previously[48]. Genome-wide DNAm profiling was conducted by the Genomics and Proteomics Core Facility of the German Cancer Research Center according to the manufacturer's protocol. In data pre-processing, signals of probes with detection P-value > 0.01, missing values >10%, and probes targeting the X and Y chromosomes were excluded. Only the CpGs that are covered by both the 450K array and EPIC array were included in the EWAS. In the KORA-Age study, data quality control and pre-processing were conducted following the CPACOR pipeline[50]. Probes with detection P-value > 0.01 and missing values >5% were removed. Quantile normalization was then performed following a coherent approach as described by Lehne et al.[50]. In addition, leukocyte composition was estimated in ESTHER and KORA-Age using the algorithms of Houseman et al.[51] for adjustment.

### Frailty assessment
Following a standard procedure[26,52], frailty in ESTHER and KORA-Age was assessed using a frailty index (FI), which is defined as the proportion of presented deficits of a predefined list of all deficits. As previously described, 31 and 33 deficits were selected for the assessment of FI in ESTHER[48] and KORA-Age[53], respectively. The lists of deficits used to define the FI in ESTHER and KORA-Age are presented in Supplementary Tables 5 and 6, respectively. Distributions of FI and proportions of the status of deficits included in the FI calculation at baseline and each follow-up in ESTHER are shown in Supplementary Fig. 6 and Supplementary Data 4, respectively. With reference to previous studies[13,54], participants were deemed frail if their FI was ≥0.250, pre-frail if their FI was >0.100 and <0.250, and non-frail if their FI was ≤0.100.

### Three-phase procedure to construct eFRS
The DNAm-based eFRS was developed using a three-phase process. A flowchart of the three-phase procedure is presented in Fig. 4. In the discovery phase, an epigenome-wide screening for frailty-related CpGs was carried out in subset I with baseline FI as a dependent variable using linear mixed regression models. The linear mixed regression models included methylation β-values as explanatory variables and adjustment for leukocyte composition as a fixed effect and batch as a random effect. After correcting for multiple testing using the Benjamini–Hochberg method[55], CpGs that reached genome-wide significance [false discovery rate (FDR) < 0.05] were validated in subset II. Similar to the discovery phase, linear mixed regression models were conducted with baseline FI as the dependent variable and additionally adjusted for age and sex. Again, CpGs with FDRs <0.05 were selected and were deemed as frailty-related loci. Then, we applied LASSO regression with a regularization parameter chosen by ten-fold cross-validation following the 'one standard error' rule to select candidates among identified CpGs and construct eFRS.

### Functional annotation of sets of CpGs
Frailty-related CpGs were annotated to genes with the information provided in the Illumina manifest file (http://emea.support.illumina.com/array/array_kits/infinium-methylationepic-beadchip-kit/downloads.html#), which is based on the University of California Santa Cruz (UCSC) and RefGene. To analyze the underlying roles of these genes, we used the Metascape online tool (https://metascape.org)[56] to perform Gene Ontology (GO) analysis, the Kyoto Encyclopedia of Genes and Genomes (KEGG) pathway analysis, and protein-protein interaction (PPI) network. Kappa scores were used as the similarity metric when performing hierarchical clustering on the enriched terms and sub-trees with a similarity of >0.3 were considered a cluster. In order to link genetic variants to variations of CpGs included in eFRS, methylation quantitative trait loci (mQTL) analysis was applied for the preliminary association analysis of single nucleotide polymorphism (SNP) sites with CpG sites. SNP-DNAm site pairs with a maximum distance of 1 Mb were tested. The analysis of mQTL was performed using the online tool mQTLdb (http://www.mqtldb.org)[57].

### Statistical analysis
Linear mixed regression and LASSO regression were performed to identify frailty-related CpGs and the eFRS in subsets I and II as aforementioned. The correlations of chronological age, eFRS, and FI at baseline and various follow-ups were assessed using Spearman correlation coefficients in subset III and subset IV. We also evaluated the correlation of eFRS and DNAm-based algorithms of aging in subset III, the AccAgeGrim (GrimAge age acceleration)[23]. The associations of eFRS with FI at baseline and various follow-ups were assessed in the two validation subsets by two linear mixed regression models that included age, sex, and leukocyte proportions as fixed effects, and batch as random effects (model 1). In further analyses, smoking status (never smoker, former smoker, current smoker) and alcohol consumption (grams per day) were additionally included as fixed effects (model 2). By categorizing baseline and various follow-up frailty statuses into two groups (non-frail versus pre-frail and frail; and non-frail and pre-frail versus frail), the associations were also estimated using a logistic regression model adjusting for all variables in model 2. In addition to logistic regression models including eFRS as a continuous variable, we also categorized eFRS according to quartiles

**Step 1. Discovery panel**
- **Data:** baseline data of ESTHER study;
- **Study population:** random samples (subset I, n=998) from ESTHER study;
- **CpGs for EWAS:** CpGs that overlapped in 450k and EPIC chips, n=422, 524 CpGs (after data pre-processing);
- **EWAS:** an epigenome-wide screening for frailty related CpGs was carried using linear mixed regression [adjusted for batch (random effect) and leukocyte composition (fixed effect)];
- **Results:** 2220 CpGs were identified (FDR < 0.05).

**Step 2. Validation panel**
- **Data:** baseline data of ESTHER study;
- **Study population:** random samples (subset II, n=730) from ESTHER study;
- **Validation:** 2220 CpGs identified from step 1 were included using linear mixed regression [adjusted for age, sex, leukocyte composition, and batch (random effect)];
- **Results:** 65 CpGs were deemed as frailty-related CpGs (FDR<0.05).

**Step 3. epigenetic frailty risk score construction**
- **Data:** baseline data of ESTHER study;
- **Construction:** LASSO regression with regularization parameter chosen by ten-fold cross-validation following the 'one standard error' rule to select CpGs for frailty risk score;
- **Results:** 20 CpGs were selected;
- **epigenetic frailty risk score** = 0.204-0.209×cg00921350-0.100×cg01234420-0.016×cg02867102-0.293×cg03725309-0.146×cg04955914-0.084×cg07312601+0.158×cg07349348+0.137×cg08463758+0.248×cg10408430-0.101×cg11700584-0.049×cg12510708+0.064×cg13570972-0.057×cg15058210-0.180×cg15380836-0.144×cg17860366+0.315×cg17971578-0.075×cg18791730-0.176×cg19267254-0.025×cg21656937-0.077×cg23458887.

**Step 4. Further validation**
**Replication in ESTHER study**
- **Data:** baseline, 2-year, 5-year, 8-year, and 11-year follow-up data
- **Study population:** random samples (subset III, n=538) from ESTHER study;

**Replication in an independent cohort: the KORA-Age study**
- **Data:** baseline, 4-year, and 8-year follow-up data
- **Study population:** random samples (subset IV, n=1010) from KORA-Age study.

**Fig. 4 | Study design and analysis flowchart.** FDR false discovery rate, EWAS epigenome-wide association study.

and run logistic regression models including eFRS as a categorical variable. FDR was also applied for multiple comparisons among results at baseline and various follow-ups and $P < 0.05$ was considered statistical significance after multiple testing.

To assess potential additional variance from passive smoking, we conducted sensitivity analyses with additional logistic regression models that controlled for smoking status using a DNAm-based proxy (the Maas 13-CpGs model)[58] rather than self-reported smoking status. Next, we assessed the association of AccAgeGrim with FI at baseline and various follow-ups.

Furthermore, we conducted subgroup analyses for the associations of the eFRS with frailty in which only non-frail participants at baseline were included for the outcome being pre-frail or frail, and only non-frail or pre-frail participants at baseline were included for the outcome being frail using model 2 for adjustment as described above.

We also systematically screened PubMed for previously reported CpGs associated with frailty and assessed the associations of these CpGs with frailty by linear-mixed regression models using model 2 for adjustment as described above.

The LASSO regression analyses were conducted using R programming (R Foundation of Statistical Computing, Vienna, Austria, version 4.0.1) package 'glmnet (version 4.1-4)'[59]. All the other statistical analyses in the ESTHER study were carried out in SAS 9.4 (SAS Institute, Cary, NC) and the analyses in the KORA-age study were conducted in R (version 4.0.1).

### Reporting summary

Further information on research design is available in the Nature Research Reporting Summary linked to this article.

### Data availability

All relevant data supporting the key findings of this study are available within the article and its supplementary information files. Due to ethical and legal restrictions, individual-level data of the two cohorts (ESTHER and KORA-age) cannot be made publicly available. Data are available upon request to X.Li (li.xiangwei@foxmail.com) and are subject to local rules and regulations. This includes submitting a proposal to the management team, where upon approval, analysis needs to be done on a local server with protected access, complying with General Data Protection Regulation. Requests will be responded to in 60 working days. Annotations of genes of frailty-related CpGs were downloaded from http://emea.support.illumina.com/array/array_kits/infinium-methylationepic-beadchip-kit/downloads.html#. The underlying roles of these genes are available at https://metascape.org[56]. Results of mQTL were downloaded from the online tool mQTLdb (http://www.mqtldb.org)[57]. The original files generated from the three websites have been deposited at https://figshare.com/articles/dataset/mQTLdb_Metascape_Annotations/20468985.

### Code availability

SAS codes for statistical analysis are available upon request.

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

## Acknowledgements

The ESTHER study was funded by grants from the Baden-Württemberg state Ministry of Science, Research and Arts (Stuttgart, Germany), the Federal Ministry of Education and Research (Berlin, Germany), the Federal Ministry of Family Affairs, Senior Citizens, Women and Youth (Berlin, Germany), and the Saarland State Ministry of Health, Social Affairs, Women and the Family (Saarbrücken, Germany). The work of Xiangwei Li was supported by a grant from Fondazione Cariplo (Bando Ricerca Malattie invecchiamento, #2017-0653). The KORA study was initiated and financed by the Helmholtz Zentrum München—German Research Center for Environmental Health, which is funded by the German Federal Ministry of Education and Research (BMBF) and by the State of Bavaria. The KORA-Age project was financed by the German Federal Ministry of Education and Research (BMBF FKZ 01ET0713 and 01ET1003A) as part of the 'Health in old age' program. Furthermore, KORA research was supported within the Munich Center of Health Sciences (MC-Health), Ludwig-Maximilians-Universität, as part of LMUinnovativ. The authors thank the study participants and their general practitioners as well as laboratory and administrative staff of the ESTHER study team. The authors gratefully acknowledge the contributions of DKFZ Genomics and Proteomics Core Facility in the processing of DNA samples and performing laboratory work. The authors thank contributions of KORA: methylation analysis and initial quality control were performed at the Core Facility Genotyping (Genome Analysis Center (GAC), Helmholtz Zentrum München) under the supervision of Nadine Lindemann, Dr. Jennifer Kriebel, and Dr. Eva Reischl. Further methylation data quality control and preprocessing were performed by Rory Wilson, Department of Molecular Epidemiology, Institute of Epidemiology, Helmholtz Zentrum München.

## Author contributions

Conception and design: X.L., H.B. Development of methodology: X.L., H.B. Acquisition of data (acquired and managed patients, provided facilities, etc.): B.H., B.S., H.B., E.G., A.P., M.W., B.T. Analysis and interpretation of data (e.g., statistical analysis, biostatistics, computational analysis): X.L., B.S., T.D. Writing of the manuscript: X.L., H.B. Critical review and revision of the manuscript: all authors Administrative, technical, or material support (i.e., reporting or organizing data, constructing databases): X.L., T.D. Study supervision: H.B.

## Funding

## Competing interests

The authors declare no competing interests.
