## [Peer Review File · Nature Communications]

Derivation and validation of an epigenetic frailty risk score in population-based cohorts of older adultsReviewers' comments:

Reviewer #1 (Remarks to the Author):

The paper by Li and colleagues considers an EWAS of frailty along with the development and application of a DNAm-based frailty risk score in the German ESTHER cohort. Overall, the paper is easy to read with analyses explained clearly.

Main comments

1. Why did you consider LASSO and not other penalised regression methods (eg elastic net)? I also don't understand why the LASSO inputs come from the EWAS model and not from the full array where there is presumably far more potential "information". With LASSO, you might also be losing information from correlated but biologically informative CpG sites.
2. The correlation matrix of the variables suggests that the frailty score is quite consistent across waves. It therefore strikes me that the results in Table 4 could be thought of as repeating similar cross-sectional analyses multiple times.
3. For the prospective prediction, I think the authors should only consider those who were healthy at baseline and go on to become frail. The FRS only improves the prediction above the null model for one follow-up wave (and not by that much) and this is without correcting for multiple testing. I don't think this provides convincing evidence of predictive utility.
4. Figure 2: I don't find the plot terribly informative. I think a scatter plot with loess curves would present the data more clearly.
5. Although the data are split into clear training and testing sets, I think replication in a completely independent cohort would strengthen the study tremendously.

Minor comments

1. Line 117: why non-parametric correlations when n is large and the data are continuous?
2. Line 122: I would include a DNAm-based proxy for smoking status instead of (or in addition) of self-reported smoking as it picks up additional variance from passive smoking and is more granular than self-report information.
3. Line 160: it would be interesting to lookup the CpGs for the FRS in the EWAS catalog to see if they have been associated with frailty-related outcomes.
4. Line 216: please quantify what you mean by "robustly". Also, please see point 3 above – I don't think the predictive performance is significant, let alone robust.
5. Line 221: please could you lookup the previously reported CpG site in your own data and report the effect size (consistent with the previous study?) and pvalue.
6. Line 272: it would be interesting to see how well GrimAge does in your models and to see if it correlates with the FRS.
7. Table 3: I might add colours/make this a heatmap to ease the interpretation. Also, age correlates with the frailty index more than the FRS, which is interesting.

Reviewer #2 (Remarks to the Author):

In this paper, Li et al. performed an EWAS and identified 65 CpGs related to a frailty index (FI), highlighting genes involving in various chronic diseases. The authors further derived a frailty risk score (FRS) based on 20 CpGs selected from penalized regression for predicting long-term frailty. Overall, I think this paper is concise and generally well-written. It also addresses an important topic, given that there has been few EWAS on frailty in the literature. However, I have several concerns as follows:

MAJOR COMMENTS

- One major limitation of this study is the absence of an external validation. I think it would be necessary to have at least one independent verification cohort to confirm the current result. It is questionable now whether these results are generalizable to other population, especially that the current sample seems to have a high proportion of low education (~75%) and overweight (~70%) individuals (Table 1). I wonder if there was an issue of selection bias here?
- Another problem is that all the C-statistics are relatively low (~0.7), and the FRS only significantly improves the predictive accuracy for FI at 5-year follow-up but not at other time points. It may therefore be an overstatement that the FRS has "robust predictive performance for frailty over long-term follow-up" (line 35).
- The authors used stratum-specific odds ratios for mortality to determine cut-off points for the FI (i.e., 0.125 and 0.30), instead of applying the more commonly used cut-off points of 0.25 or 0.21 in the literature. While this method could identify frail individuals with the highest mortality risk in the current sample, I wonder if this will make it harder to compare the results with other studies? Besides, as the authors have noted, the FI was constructed based on a large number of self-reported items related to history of diseases and drug use (Supplementary Table 1) which may lead to potential reporting bias. It would be helpful if the authors can also provide information regarding distribution of the FI.
- The authors did not provide any relevant functional annotations, such as pathway/network analyses for the genes that the significant hits harbor. I would also expect to see some other complementary analyses, such as assessing whether the identified sites are genetically regulated (e.g. association with mQTL).
- More details regarding sample selection are needed. What are the inclusion and exclusion criteria? What are the proportions of missing data for the FI and FRS? It is also unclear how the three subsets were selected and how the sample sizes of 998, 730, and 538 were obtained (line 66)?

MINOR COMMENTS

- How much in variation of frailty does the FRS explain?
- I found the name "frailty risk score" (FRS) a bit confusing, given that some other measures with similar names such as the Hospital Frailty Risk Score are already widely used. The authors may perhaps consider using a more specific name, such as something like the "epigenetic frailty risk score"?
- It is better to use the term area under the curve (AUC) instead of C-statistics.
- Should BMI and education be also adjusted in the models?
- I would prefer mentioning the currently available frailty measures in Introduction instead of in the middle of Discussion (lines 246-252).
- The authors mentioned that there is one CpG overlap (i.e., cg05575921) between the FRS and the MRscore (line 273), but I was not able to find this CpG in the Tables or in the FRS equation. Please check.
- Figure 1: the sample sizes of the three subsets (1030, 730, 548) were different from that written in Methods (998, 730, 538). Please check.
- Table 4: It is a bit difficult to understand what the estimates refer to. I would prefer putting Supplementary Table 4 (showing ORs for the associations between FRS and being frail) in the main text instead.
- Figure 2: Seems that there is an error in the y-axis, in which "0.20" appeared two times.

Responses to the reviewers' comments

Reviewer #1 (Remarks to the Author):

The paper by Li and colleagues considers an EWAS of frailty along with the development and application of a DNAm-based frailty risk score in the German ESTHER cohort. Overall, the paper is easy to read with analyses explained clearly.

Response: Thank you very much for the appreciation of our work!

Main comments

1. Why did you consider LASSO and not other penalised regression methods (eg elastic net)? I also don't understand why the LASSO inputs come from the EWAS model and not from the full array where there is presumably far more potential "information". With LASSO, you might also be losing information from correlated but biologically informative CpG sites.

Response: We also considered and applied elastic net, but the resulting algorithm was slightly less predictive of FI in the validation subset (subset III) despite including a slightly higher number of CpGs (25 CpGs). Given the large number of highly correlated CpGs in the ILLUMINA methylation arrays, pre-filtering steps are a commonly applied reasonable first step in their analysis. In our analysis we followed an EWAS approach for filtering CpGs that has previously successfully applied for deriving a very informative, validated and widely cited score for predicting mortality (Zhang et al, Nat Commun. 2017;8:14617).

2. The correlation matrix of the variables suggests that the frailty score is quite consistent across waves. It therefore strikes me that the results in Table 4 could be thought of as repeating similar cross-sectional analyses multiple times.

Response: Yes, there is high, but far from perfect correlation of the FI between various waves, with correlations gradually decreasing with increasing time intervals as expected. The degree of correlation and its impact on predictiveness of eFRS according to time interval is one of the findings of this study.

3. For the prospective prediction, I think the authors should only consider those who were healthy at baseline and go on to become frail. The FRS only improves the prediction above the null model for one follow-up wave (and not by that much) and this is without correcting for multiple testing. I don't think this provides convincing evidence of predictive utility.

Response: We now conducted additional analyses for prospective prediction in which only non-frail participants at baseline were included for the outcome "pre-frail or frail", and only non-frail or pre-frail participants at baseline were included for the outcome "frail". Overall, very similar results were obtained (new Supplementary Figures 3 and 4; Page 11, Line 205, Page 12 Lines 206-208; Page 17, Lines 320-321, Page 18, Lines 322-323; and Page 18, Lines 333-335).

4. Figure 2: I don't find the plot terribly informative. I think a scatter plot with loess curves would present the data more clearly.

Response: Thank you for spotting this redundancy. We have removed previous Figure 2.

5. Although the data are split into clear training and testing sets, I think replication in a completely independent cohort would strengthen the study tremendously.

Response: Thank you for addressing this very important point. We now carried out a replication study in a completely independent cohort, the KORA-Age study, and this validation study fully confirmed our initial results, which indeed tremendously strengthened the study.

Minor comments

1. Line 117: why non-parametric correlations when n is large and the data are continuous?

Response: We used Spearman correlations because the FIs are not normally distributed variables (Shapiro-Wilk tests with P values < 0.05).

2. Line 122: I would include a DNAm-based proxy for smoking status instead of (or in addition) of self-reported smoking as it picks up additional variance from passive smoking and is more granular than self-report information.

Response: We now carried out sensitivity analyses with models adjusting for smoking status by a DNAm-based model (Eur J Epidemiol. 2019;34(11):1055-74, the Mass 13-CpGs model) rather than self-report smoking status. Results were very similar for both types of models (new Supplementary Table 8; Page 11, Lines 200-203 and Page 18, Lines 336-339).

3. Line 160: it would be interesting to lookup the CpGs for the FRS in the EWAS catalog to see if they have been associated with frailty-related outcomes.

Response: We addressed this important point by adding the new Supplementary Table 6 and pertinent text (Page 15, Lines 257-262).

4. Line 216: please quantify what you mean by "robustly". Also, please see point 3 above – I don't think the predictive performance is significant, let alone robust.

Response: We have revised the sentence and deleted the word "robustly" (Page 20, Line 370).

5. Line 221: please could you lookup the previously reported CpG site in your own data and report the effect size (consistent with the previous study?) and p-value.

Response: We systematically screened previous studies and identified 15 CpGs from 3 studies. Eight of them showed statistically significant associations with frailty in our cohort (new Supplementary Table 5; Page 12, Lines 209-211 and Page 14, Lines 241-243).

6. Line 272: it would be interesting to see how well GrimAge does in your models and to see if it correlates with the FRS.

Response: We now report the correlation between eFRS and GrimAge (rSp = 0.566, Page 16, Lines 289-290). In addition, we now have assessed the associations of GrimAge with FI at baseline and follow-ups (new Supplementary Table 9, Supplementary Figures 5 and 6; Page 11, Lines 203-204, Page 19, Lines 351-357).

7. Table 3: I might add colours/make this a heatmap to ease the interpretation. Also, age correlates with the frailty index more than the FRS, which is interesting.

Response: Revised accordingly (Figure 2).

Reviewer #2 (Remarks to the Author):

In this paper, Li et al. performed an EWAS and identified 65 CpGs related to a frailty index (FI), highlighting genes involving in various chronic diseases. The authors further derived a frailty risk score (FRS) based on 20 CpGs selected from penalized regression for predicting long-term frailty. Overall, I think this paper is concise and generally well-written. It also addresses an important topic, given that there has been few EWAS on frailty in the literature.

Response: Thank you very much for the appreciation of our work!

However, I have several concerns as follows:

MAJOR COMMENTS

1. One major limitation of this study is the absence of an external validation. I think it would be necessary to have at least one independent verification cohort to confirm the current result.

Response: Thank you very much for this important suggestion! We now carried out a replication study in a completely independent cohort, the KORA-Age study, and this validation study fully confirmed our initial results.

2. It is questionable now whether these results are generalizable to other population, especially that the current sample seems to have a high proportion of low education (~75%) and overweight (~70%) individuals (Table 1). I wonder if there was an issue of selection bias here?

Response: As previously reported (new reference 28), the ESTHER cohort was found to be representative of the older German population with respect to major sociodemographic, lifestyle and medical characteristics. We now have added this important information on Page 6, Lines 85-90.

3. Another problem is that all the C-statistics are relatively low (~0.7), and the FRS only significantly improves the predictive accuracy for FI at 5-year follow-up but not at other time points. It may therefore be an overstatement that the FRS has “robust predictive performance for frailty over long-term follow-up” (line 35).

Response: We have reworded the respective sentences to avoid overstatement (Page 3, Line 50 and Page 20, Line 370).

4. The authors used stratum-specific odds ratios for mortality to determine cut-off points for the FI (i.e., 0.125 and 0.30), instead of applying the more commonly used cut-off points of 0.25 or 0.21 in the literature. While this method could identify frail individuals with the highest mortality risk in the current sample, I wonder if this will make it harder to compare the results with other studies? Besides, as the authors have noted, the FI was constructed based on a large number of self-reported items related to history of diseases and drug use (Supplementary Table 1) which may lead to potential reporting bias. It would be helpful if the authors can also provide information regarding distribution of the FI.

Response: Thank you for the constructive suggestion which we were happy to follow. We now have used commonly used cut-off points (0.100 and 0.250). We now added detailed information on distribution of the FI and its single components in the new Supplementary Figure 1 and Supplementary Table 3.

5. The authors did not provide any relevant functional annotations, such as pathway/network analyses for the genes that the significant hits harbor. I would also expect to see some other complementary analyses, such as assessing whether the identified sites are genetically regulated (e.g. association with mQTL).

Response: Thank you for addressing this important point! We now provide the results of pathway/network analyses for the genes in the new Supplementary Figure 2 and added pertinent text (Page 10, Lines 169-177 and Page 15, Lines 263-272). Information on mQTL analyses is presented in the new Supplementary Table 7 and pertinent text (Page 10, Lines 177-182 and Page 15, Lines 273-276).

6. More details regarding sample selection are needed. What are the inclusion and exclusion criteria? What are the proportions of missing data for the FI and FRS? It is also unclear how the three subsets were selected and how the sample sizes of 998, 730, and 538 were obtained (line 66)?

Response: We added more details of sample selection in the Methods section (Page 6, Lines 97-99 and 21, Page 7, Lines 100-103).

MINOR COMMENTS

1. How much in variation of frailty does the FRS explain?

Response: We now added this information as requested (Page 16, Lines 287-289 and Lines 294-295).

2. I found the name “frailty risk score” (FRS) a bit confusing, given that some other measures with similar names such as the Hospital Frailty Risk Score are already widely used. The authors may perhaps consider using a more specific name, such as something like the “epigenetic frailty risk score”?

Response: We fully agree with the reviewer’s suggestion and now more precisely

refer to ‘frailty risk score’ as ‘epigenetic frailty risk score’ throughout the paper.

3. It is better to use the term area under the curve (AUC) instead of C-statistics.

Response: We now use the term area under the curve instead of C-statistics.

4. Should BMI and education be also adjusted in the models?

Response: The following table shows results on regression models with and without BMI and education level. All of the associations are highly consistent with such additional adjustment.

Table. Association of eFRS with being pre-frail or frail by regression models adjusted for different factors

	OR (95% CI, per SD of eFRS)	
	Models without education level and BMI ^a	Models with education level and BMI ^b
Baseline	1.38 (1.05-1.82)	1.36 (1.03-1.80)
2-year follow-up	1.67 (1.26-2.23)	1.64 (1.22-2.20)
5-year follow-up	1.38 (1.01-1.87)	1.39 (1.02-1.87)
8-year follow-up	1.39 (0.96-2.02)	1.40 (0.96-2.05)
11-year follow-up	0.93 (0.59-1.47)	0.94 (0.59-1.49)

Abbreviations: eFRS, epigenetic frailty risk score; OR, odds ratio; CI, confidence interval; SD, standard deviation; BMI, body mass index.

^a Models were adjusted for age, sex, leukocyte composition, batch, baseline smoking status (never smoker, former smoker, current smoker), and alcohol consumption (grams per day).

^b Models were adjusted for age, sex, leukocyte composition, batch, baseline smoking status (never smoker, former smoker, current smoker), and alcohol consumption (grams per day), education level (≤ 9 years, 10 - 11 years, ≥ 12 years), BMI (kg/m²).

5. I would prefer mentioning the currently available frailty measures in Introduction instead of in the middle of Discussion (lines 246-252).

Response: We now moved the description of currently available frailty measures to the Introduction (Page 4, Lines 60-68).

6. The authors mentioned that there is one CpG overlap (i.e., cg05575921) between the FRS and the MRscore (line 273), but I was not able to find this CpG in the Tables or in the FRS equation. Please check.

Response: We apologize for this apparent mistake which we now have corrected.

7. Figure 1: the sample sizes of the three subsets (1030, 730, 548) were different from that written in Methods (998, 730, 538). Please check.

Response: We apologize for this apparent slight inconsistency. We have subsequently carefully rechecked consistency of numbers reported in tables, figures and text.

8. Table 4: It is a bit difficult to understand what the estimates refer to. I would prefer putting Supplementary Table 4 (showing ORs for the associations between FRS and being frail) in the main text instead.

Response: The previous Supplementary Table 4 has been moved to the main text (new Figure 4).

9. Figure 2: Seems that there is an error in the y-axis, in which “0.20” appeared two times.

Response: Previous Figure 2 has been removed.

REVIEWER COMMENTS

Reviewer #1 (Remarks to the Author):

I thank the authors for their responses to my queries. For the most part, I am satisfied with their edits. However, I have a few remaining comments.

1. I think there is a degree of redundancy over lines 203 to 323. Table 3 shows the associations between the eFRS and FI while Figure 3 shows the associations between the eFRS and frailty categories. I appreciate that in the former you are looking at a continuous variable before moving to a categorical in the latter. However, I would maybe report one and have the other in a supplementary file.

2. I think the results in Supplementary Figures 3 and 4 are far more informative than those in Figures 3 and 4. The association between baseline eFRS and follow-up frailty is only really interesting (I think) for those who were not frail at baseline. Currently, I find the results quite difficult to interpret as the same analyses (broadly speaking) seem to be reported multiple times in slightly different formats e.g., (1) eFRS and FI at baseline without covariates, (2) eFRS and FI at baseline with covariates, (3) eFRS and pre-frail/frail at baseline logistic model, (4) eFRS and frail at baseline logistic model, (4) eFRS and frail (y/n) ROC curve. I wonder if streamlining the results would improve the narrative and make the paper easier for the reader to follow.

3. The GrimAge results in Supplementary Figure S6 do show a significant effect at the 5y follow-up but the text states there were only null findings. The authors report quite a high correlation between GrimAge and the eFRS ($r=0.57$). Do you know what aspect of GrimAge is driving the association? It would be interesting to look at how its component parts associate with the eFRS.

4. What strategy was take to account for multiple testing? I asked this in my previous review but it has not been addressed.

Reviewer #2 (Remarks to the Author):

I would like to thank the authors for the thoughtful response and the additional comprehensive analyses. The manuscript is greatly improved by inclusion of the external validation cohort and the sensitivity analyses. I have no further comments.

Responses to the reviewers' comments

Reviewer #1 (Remarks to the Author):

I thank the authors for their responses to my queries. For the most part, I am satisfied with their edits. However, I have a few remaining comments.

1. I think there is a degree of redundancy over lines 203 to 323. Table 3 shows the associations between the eFRS and FI while Figure 3 shows the associations between the eFRS and frailty categories. I appreciate that in the former you are looking at a continuous variable before moving to a categorical in the latter. However, I would maybe report one and have the other in a supplementary file.

Response: Thank you for spotting this redundancy. We have moved previous Table 3 to appendix (new Supplementary Table 8).

2. I think the results in Supplementary Figures 3 and 4 are far more informative than those in Figures 3 and 4. The association between baseline eFRS and follow-up frailty is only really interesting (I think) for those who were not frail at baseline. Currently, I find the results quite difficult to interpret as the same analyses (broadly speaking) seem to be reported multiple times in slightly different formats e.g., (1) eFRS and FI at baseline without covariates, (2) eFRS and FI at baseline with covariates, (3) eFRS and pre-frail/frail at baseline logistic model, (4) eFRS and frail at baseline logistic model, (4) eFRS and frail (y/n) ROC curve. I wonder if streamlining the results would improve the narrative and make the paper easier for the reader to follow.

Response: Thank you for pointing this out. We now have moved results on the associations of eFRS with FI and being pre-frail/frail to the appendix (previous Figure 3 and Table 3). Meanwhile, we moved previous Supplementary Figure 4 to the main text as new Figure 4. Finally, only results on the association of eFRS with being frail and ROC curve of eFRS and being frail are reported in the main text.

3. The GrimAge results in Supplementary Figure S6 do show a significant effect at the 5y follow-up but the text states there were only null findings. The authors report quite a high correlation between GrimAge and the eFRS ($r=0.57$). Do you know what aspect of GrimAge is driving the association? It would be interesting to look at how its component parts associate with the eFRS.

Response: We apologize for this apparent slight inconsistency and have subsequently carefully rechecked consistency of all results reported in tables, figures and text.

We now discussed the potential reason of the high correlation between GrimAge and eFRS in Discussion section (Page 22, Lines 14-18).

4. What strategy was take to account for multiple testing? I asked this in my previous review but it has not been addressed.

Response: We now clarified the strategy for correcting for multiple testing in

Methods section (Page 11, Lines 17-19).

Reviewer #2 (Remarks to the Author):

I would like to thank the authors for the thoughtful response and the additional comprehensive analyses. The manuscript is greatly improved by inclusion of the external validation cohort and the sensitivity analyses. I have no further comments.

Response: Thank you very much for the appreciation of our work!